# Coalescence of carbon nanotubes while preserving the chiral angles

Akira Takakura [1], Taishi Nishihara [1], Koji Harano [2,3], Ovidiu Cretu[2], Takeshi Tanaka [4], Hiromichi Kataura [4] & Yuhei Miyauchi [1] ✉

Atomically precise coalescence of graphitic nanocarbon molecules is one of the most challenging reactions in $sp^2$ carbon chemistry. Here, we demonstrate that two carbon nanotubes with the same chiral indices $(n, m)$ are efficiently coalesced into a single $(2n, 2m)$ nanotube with preserved chiral angles via heat treatment at less than 1000 °C. The $(2n, 2m)$ nanotubes constitute up to ≈ 20%–40% of the final sample in the most efficient case. Additional optical absorption peaks of the $(2n, 2m)$ nanotubes emerge, indicating that the reaction occurs over the entire sample. The reaction efficiency strongly depends on the chiral angle, implying that C–C bond cleavage and recombination occurs sequentially. Furthermore, the reaction occurs efficiently even at 600 °C in an atmosphere containing trace amounts of oxygen. These findings offer routes for the structure-selective synthesis of large-diameter nanotubes and modification of the properties of nanotube assemblies via postprocessing.

Joining or fusing graphitic nanocarbon materials comprising a hexagonal $sp^2$ carbon network in a well-controlled manner is among the most challenging issues in $sp^2$ carbon chemistry as it requires the cleavage of numerous C–C bonds and recombination into $sp^2$ bonds with atomic precision. The conversion of large nanocarbon macromolecules such as fullerenes and carbon nanotubes into large, coalesced molecules[1–9] is a well-known conundrum (Fig. 1a, b). Since the electronic and optical properties of these nanocarbon materials are dominated by the geometry and topology of $\pi$ electrons delocalized on the $sp^2$ carbon network[10], the coalescence of these nanocarbon materials can drastically modify their properties. The seamless coalescence of two single-walled carbon nanotubes (hereafter referred to as nanotubes) into one thick nanotube is of particular interest (Fig. 1b). Because electronic[11–13], optical[14,15], chemical[16,17], thermal[18,19], and mechanical[20,21] properties of nanotubes strongly depend on their structure defined by their diameter and chiral angle (or chiral indices $(n, m)$) (Fig. 1c), considerable effort has been devoted to achieving fully structure-controlled synthesis[22,23] or structural separation[24–31] of nanotubes. However, structure-controlled synthesis or separation methods have remained limited to $(n, m)$ species with relatively small diameters (≈ 1 nm or less), and precisely structure-selective synthesis or sorting of thick nanotubes with diameters of more than ≈ 1.3 nm has not been realized because of the great variety of geometrically possible structures of nanotubes with similar diameters and properties. Atomically precise nanotube coalescence may address this issue. If the efficient coalescence reaction of small-diameter $(n, m)$ nanotubes for which high-purity samples are available[24–31] can be triggered, structure-selective synthesis of thick $(2n, 2m)$ nanotubes can be achieved from the $(n, m)$ nanotubes as precursors (Fig. 1b). Furthermore, the macroscopic properties of bulk nanotube aggregates, including their electronic and thermal conductivities, can be dramatically modified by forming covalent bonds between carbon nanotubes in the aggregate using partial coalescence reactions, and may provide alternative ways for modifying the properties of nanotube assemblies via postprocessing.

However, the coalescence of nanotubes remains challenging. Although the connection of two fullerene molecules via solid-state reaction[2] and the conversion of fullerene peapods into double-walled

[1]Institute of Advanced Energy, Kyoto University, Uji, Kyoto 611-0011, Japan. [2]Center for Basic Research on Materials, National Institute for Materials Science, Tsukuba, Ibaraki 305-0044, Japan. [3]Research Center for Autonomous Systems Materialogy (ASMat), Institute of Integrated Research, Institute of Science Tokyo, Yokohama, Kanagawa 226–8501, Japan. [4]Nanomaterials Research Institute, AIST, Tsukuba, Ibaraki 305-8565, Japan. ✉ e-mail: miyauchi@iae.kyoto-u.ac.jp

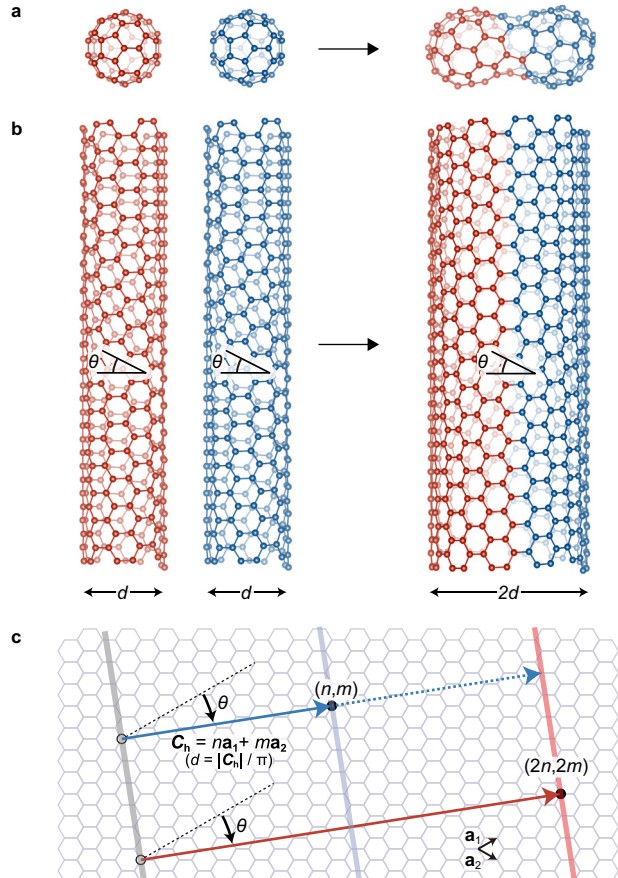

**Fig. 1 | Coalescence of nanocarbon molecules.** Coalescence of fullerenes (**a**) and carbon nanotubes (**b**) into large coalesced molecules. $\theta$ and $d$ in (**b**) are the chiral angle and diameter, respectively. (**c**) Structure of carbon nanotubes specified by the chiral indices $(n, m)$, which represent the coordinates of the chiral vector (blue solid arrow) with respect to the basis $\mathbf{a_1}$ and $\mathbf{a_2}$. The chiral vector ($\mathbf{C_h}$) connects two carbon atoms (open and filled circles) on a graphene plane, and a nanotube is formed by rolling to connect them (gray and blue solid lines). The diameter $d$ is described using $|\mathbf{C_h}|/\pi$. Geometrically, if two nanotubes with identical $(n, m)$ structures (the chiral vectors are indicated by the blue solid and dotted arrows) are coalesced into one thick nanotube, the chiral angle $\theta$ of the nanotubes before and after coalescence can be preserved, and the diameter $d$ doubles (chiral vector indicated as a red arrow)[4,36]; thus, the chiral indices $(n, m)$ after the coalescence reaction are expected to be $(2n, 2m)$. In (**b**), the coalescence reaction of two (6,5) nanotubes into a (12,10) nanotube is shown.

carbon nanotubes at temperatures higher than 1200 °C[6] have been reported, the number of $sp^2$ C–C bonds that must be cleaved and reconstructed for the nanotube coalescence is much larger than that in fullerenes. In addition, the chemical and thermal stability of nanotubes is much higher than that of fullerenes. Previous studies have shown that partial removal of carbon atoms by electron irradiation at 800 °C in transmission electron microscopy (TEM) could initiate the coalescence reaction of nanotubes[4]. The existence of small amounts of coalesced nanotubes after heat treatment of nanotube aggregations at more than 1700 °C was also found by TEM observations[5]. However, the realization of efficient coalescence all over the macroscopic nanotube aggregations remains an open issue.

Here, we report efficient coalescence of $(n, m)$ carbon nanotubes into doubled $(2n, 2m)$ nanotubes enabled using heat treatment. After heat treatment at 900–1000 °C under low pressure, micro-Raman spectroscopy measurements and aberration-corrected TEM observations were used to verify the chiral-angle-preserving coalescence of $(n,$

$m)$ nanotubes into $(2n, 2m)$ nanotubes. Optical absorption peaks of the exciton resonance for the $(2n, 2m)$ structures emerged after the reaction, whereas the absorption intensity of the original $(n, m)$ structures decreased. Moreover, the efficiency of the coalescence strongly depended on the chiral angle of the nanotubes; only armchair $n = m$ ($\theta = 30°$) and near-armchair $n \approx m$ ($\theta \approx 30°$) types showed efficient coalescence. The content of coalescence-derived $(2n, 2m)$ nanotubes in the final product reached 20%–40% for the armchair and near-armchair cases, as evaluated using optical absorption spectroscopy. In contrast, near-zigzag $(n, m)$ nanotubes with $n >> m$ showed low coalescence efficiency. These results can be explained based on the chirality dependence of energy costs for the coalescence reaction originating from geometric factors. Furthermore, we found that the coalescence reaction occurs efficiently even at 600 °C when trace amounts of oxygen is introduced into the reaction chamber. These findings offer an approach for synthesizing chiral-structure-controlled carbon nanotubes with large diameters as well as an opportunity to modify the physical properties of carbon nanotube aggregates using postprocessing, which may be useful for their use as bulk materials in various applications.

## Results and discussion
### Coalescence reaction of (6,5) nanotubes
First, we demonstrate that efficient coalescence of $(n, m) = (6,5)$ near-armchair carbon nanotubes into doubled $(2n, 2m) = (12,10)$ nanotubes is feasible. One possible reason for the low efficiency of the coalescence reaction in previous studies may be the use of mixed nanotubes with various chiral structures specified by either the chiral indices $(n, m)$ or the chiral angle and diameter $(\theta, d)$ (Fig. 1c), which hardly coalesce geometrically. Therefore, in this study, we examined the coalescence reaction using the aggregation of carbon nanotubes with the same $(n, m)$ structures. We fabricated nanotube membranes via the filtration of (6,5)-enriched dispersions[32] (Fig. 2a) prepared using the gel chromatography separation method[29]. The membrane was transferred onto the sapphire substrate (Fig. 2b), followed by heat treatment in a vacuum using an electric furnace (see Methods for the details of the sample preparation and the heating experiment). Figure 2c, d shows the optical absorption of (6,5)-enriched nanotube membranes before and after heat treatment at 1000 °C, measured using optical transmission spectroscopy (see Supplementary Fig. 1 for the broadband spectra up to 3 eV). In these measurements, the optical absorption was measured using an incident light beam covering the entire substrate. Thus, the absorption spectra represent the ensemble-averaged optical properties of the membranes. The strong absorption peak arising from the first sub-band exciton ($S_{11}$) for (6,5) nanotubes was observed at 1.21 eV before and after heat treatment. After heat treatment, the intensity of this exciton peak decreased, indicating the decreased number of (6,5) nanotubes in the membrane. In addition, a distinct absorption peak emerged at 0.67 eV after the heat treatment (the asterisk in Fig. 2d). According to the table of the excitation photon energy for each $(n, m)$ nanotube type, namely the Kataura plot[14,33], the $S_{11}$ exciton absorption peak energies of doubled $(2n, 2m)$ species are expected to be approximately half of those of the $(n, m)$ nanotubes, while their $S_{22}$ exciton absorption peak energies should be nearly the same as the $S_{11}$ peak energies of the original $(n, m)$ species. Thus, the emergence of this additional low energy peak originating from the exciton resonance of nanotubes thicker than the original (6,5) species is the signature of efficient nanotube coalescence reaction throughout the membrane. The reaction occurred in the entire sample is also supported by the macroscopic changes in transport properties before and after the nanotube coalescence reaction (see Methods and Supplementary Fig. 2). The insets in Fig. 2c, d show typical radial-breathing-mode (RBM) Raman spectra of the sample before and after heat treatment, respectively. Since the RBM Raman shift is inversely proportional to the nanotube diameter[34], the appearance of an

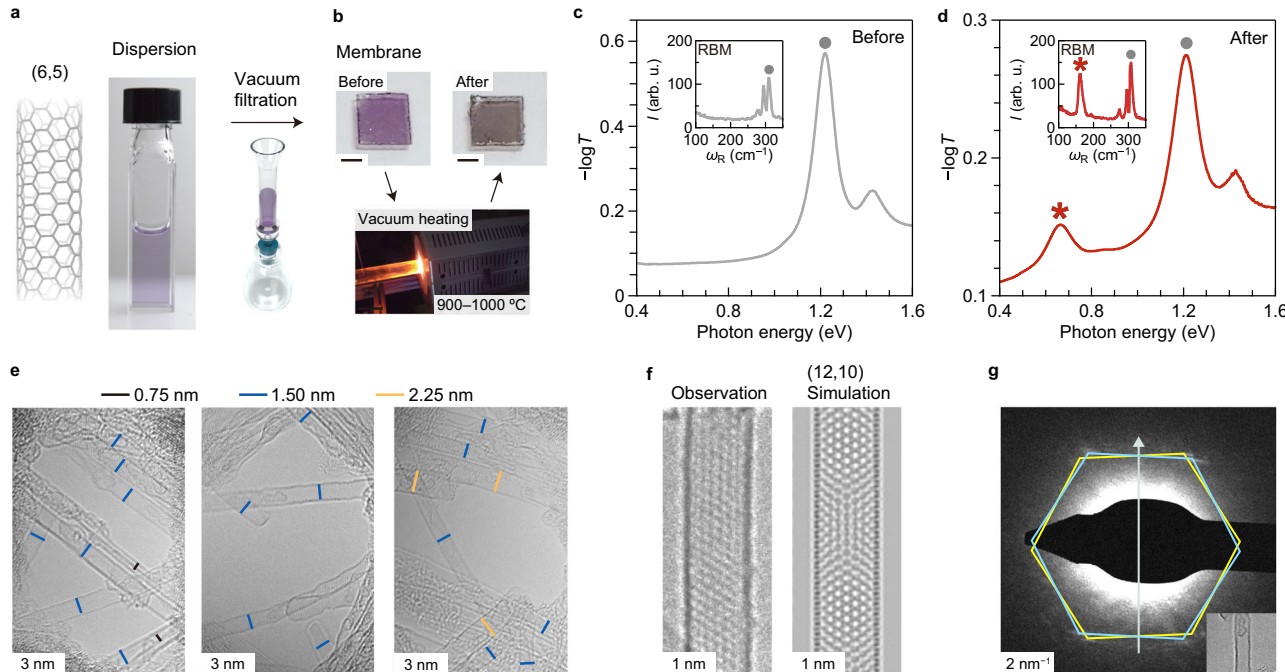

**Fig. 2 | Evidence of efficient carbon nanotube coalescence with preserved chiral angles. a** Schematic of the (6,5) carbon nanotube (left) and a photograph of its dispersion (right). **b** Photographs of the (6,5) nanotube membrane before and after vacuum heat treatment at 1000 °C. Scale bar, 2 mm. Optical absorption ($-\log T$, where $T$ is transmittance) spectra of (6,5) nanotube membranes before (**c**) and after (**d**) the vacuum heat treatment. The insets show the RBM features of the Raman spectra. $I$ and $\omega_R$ are intensity and Raman shift. The gray filled circles indicate the peaks corresponding to the (6,5) nanotubes. **e** Transmission electron microscope (TEM) images obtained for the samples after vacuum heat treatment at 900 °C. Bars

with different colors indicates lengths corresponding to diameters of typical nanotubes found in the sample. **f** Comparison of the observed TEM image of the generated nanotube with $d \approx 1.5$ nm and simulated one for a (12,10) nanotube. **g** A nanobeam diffraction pattern for the newly generated nanotube with $d \approx 1.5$ nm. Two sets of diffraction spots are positioned at the apices of regular hexagons (yellow and blue). The gray arrow indicates the direction of the CNT axis. Inset indicates the real space TEM image of the observed nanotube. Scale bar, 2 nm. Source data are provided as a Source Data file.

additional RBM peak (indicated by an asterisk) at approximately half the RBM Raman shift of the (6,5) nanotube (indicated by a gray-filled circle) also supports the formation of nanotubes with twice the diameter.

To confirm that the emergence of the additional absorption peak was due to the nanotube coalescence more directly, we performed TEM observations using aberration-corrected TEM on thin (6,5) membranes before and after heat treatment (see Methods for details). Because the nanotube membranes prepared for optical transmission measurements were too thick, we prepared thinner membranes for TEM observations. After heat treatment, some parts of the membranes were broken and missing, but we managed to find isolated or small bundled nanotubes that allowed TEM observations. In the representative TEM images of the (6,5) nanotube membrane before heat treatment, we observed only small-diameter ($d \approx 0.75$ nm) nanotubes (Supplementary Fig. 3a). Although some damage was found on the nanotube sidewalls presumably because of the electron beam irradiation during the TEM observation, the observed diameter ($d \approx 0.75$ nm) and chiral angle ($\theta \approx 27°$) were consistent with those of (6,5). In contrast, in the membrane after heat treatment (Fig. 2e), we observed many nanotubes with diameters of $\approx 1.5$ nm that is nearly twice the diameter of the (6,5) nanotubes. Figure 2f compares the observed high-resolution TEM image of a nanotube formed after heat treatment with a diameter of $\approx 1.5$ nm to the simulated TEM image of a (12,10) nanotube (see Methods for details of the simulation). The moiré pattern, caused by the interference of the hexagonal carbon network textures on the top and bottom surfaces of the nanotube, is consistent with that expected for a (12,10) nanotube. We also confirmed the chiral angle $\theta \approx 27°$ through the nanobeam diffraction pattern as shown in Fig. 2g (see Methods for details on the diffraction measurements and

chiral angle determination). These observations indicate that the original (6,5) nanotubes coalesced, with the original chiral angle preserved after the coalescence of the nanotubes. We also observed a small number of thicker nanotubes with nearly triple the diameter (Fig. 2e and Supplementary Fig. 3b), suggesting that the coalescence of three semiconducting (6,5) nanotubes into a single nanotube with triple the diameter can also occur.

## Chirality dependence
Further, we studied the chirality dependence of the nanotube coalescence reaction efficiency. We prepared single-chirality nanotube membranes in which the major components were (10,0), (9,1), (9,2), (8,3), (6,5), and (6,6) (see Methods). Among them, only (6,6) was the metallic type, and the others were semiconducting. These nanotubes had similar diameters ($\approx 0.7$–$0.8$ nm) with different chiral angles, as shown in Fig. 3a. Therefore, we focused on the dependence of the coalescence efficiency on the chiral angle in this experiment. As shown in the photographs in Fig. 3a, the nanotube dispersions exhibited a variety of colors, reflecting the chirality-dependent resonance energies of the second sub-band excitons[32,35]. Figure 3b shows the optical absorption of each $(n, m)$ membrane after heat treatment at 1000 °C (those before heat treatment are shown in Supplementary Fig. 4). The emergence of distinct absorption peaks (indicated by the asterisks) at $\approx 0.5$–$0.6$ times the photon energy of the first sub-band exciton resonance (indicated by the filled circles) was observed for near-armchair (6,5) and armchair (6,6) species, both with large chiral angles of $\approx 30°$. These results are in stark contrast to those of the near-zigzag and zigzag species. Only a very small peak [(8,3), (9,2)] or almost no peak [(10,0), (9,1)] was observed for these nanotube types with chiral angles of $\approx 0°$. Figure 3c shows the coalescence efficiency as a function of

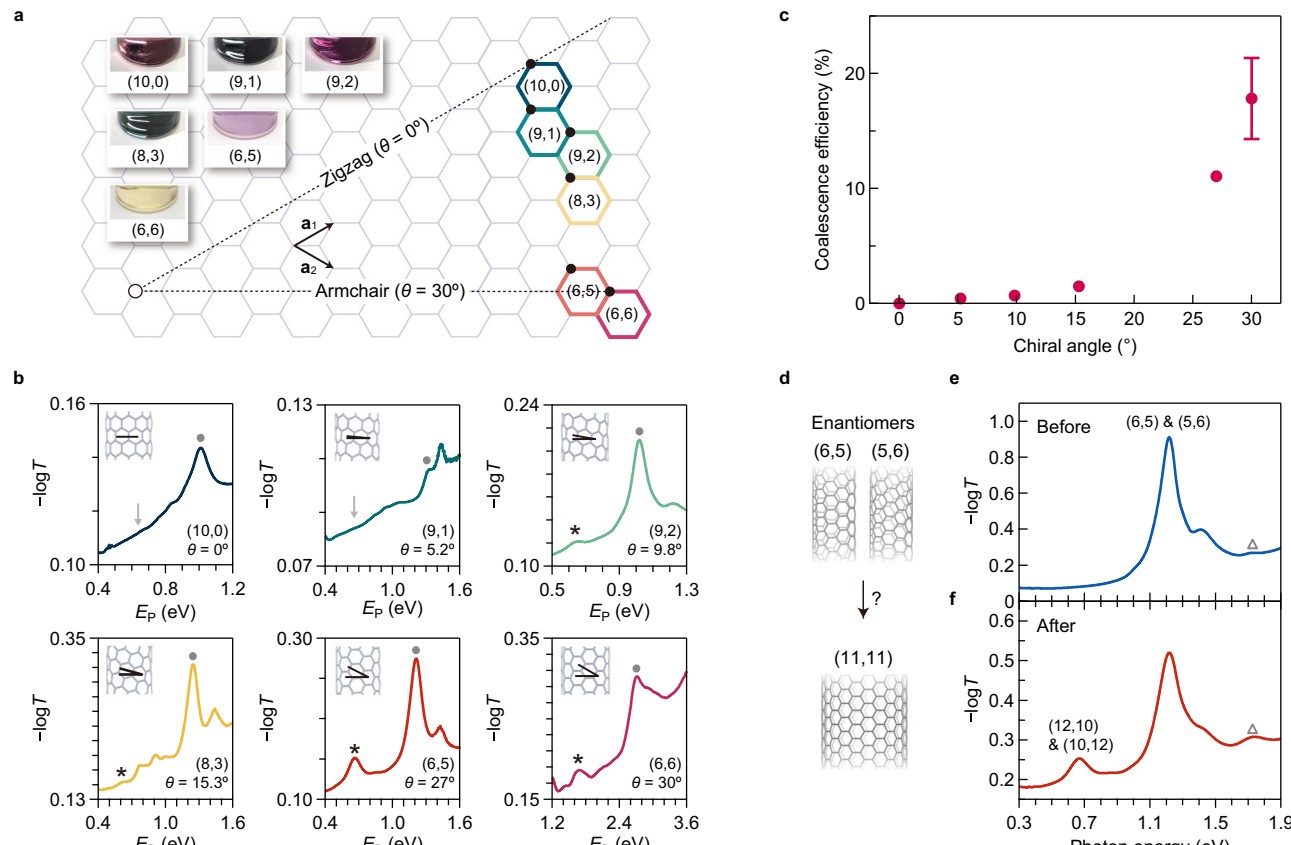

**Fig. 3 | Chirality dependence of the coalescence reaction. a** Map of $(n, m)$ nanotubes examined in this study with optical images of the dispersions of each $(n, m)$ carbon nanotube. **b** Optical absorption ($-\log T$, where $T$ is transmittance) spectra of the membranes after vacuum heat treatment at 1000 °C (the one for (6,5) is the same as that in Fig. 2d). $E_P$, photon energy. The filled circles and asterisks indicate the lowest energy exciton peaks of $(n, m)$ and $(2n, 2m)$ nanotubes, respectively. The solid arrows indicate the photon energy of the lowest energy exciton peak of $(2n, 2m)$ nanotubes, but no peak appears. The insets show the schematics of nanotubes with two lines showing the chiral angles. **c** Efficiency of the coalescence reaction as

a function of chiral angle. The error bar for the data at $\theta = 30°$ represents the standard error obtained by performing multi-peak fitting to the spectral data of a single representative sample, using Igor Pro® software. These errors for other chiral angles are negligible and not shown. **d** Schematic depicting the coalescence of the enantiomers of (6,5) and (5,6) nanotubes. Optical absorption spectrum of (6,5) and (5,6) nanotube membranes before (**e**) and after (**f**) heat treatment. The open triangles indicate the expected photon energy corresponding to the first sub-band exciton of (11,11) nanotubes. Source data are provided as a Source Data file.

chiral angle. The efficiency for each $(n, m)$ type, $\eta_{nm}$, was calculated using the integrated peak area of the original $S_{11}$ of $(n, m)$ nanotubes before heat treatment ($A_{nm}^{\text{before}}$) and that of emerged $(2n, 2m)$ nanotubes after heat treatment ($A_{2n2m}$) (see Methods and Supplementary Figs. 5 and 6 for details) as $\eta_{nm} = A_{2n2m}/A_{nm}^{\text{before}}$. Under the current experimental conditions, some portions of the nanotubes were lost after heat treatment, but the remaining nanotubes were converted into the doubled $(2n, 2m)$ species. For the (near) armchair (6,5) and (6,6) nanotubes, the efficiency reached ≈ 10%–20%. Regarding the content ratio of the emerged $(2n, 2m)$ species in the final sample, defined as $A_{2n2m}/(A_{nm}^{\text{after}} + A_{2n2m})$, where $A_{nm}^{\text{after}}$ is the integrated peak areas of the original $S_{11}$ of $(n, m)$ nanotubes after heat treatment, they were ≈ 20%–40% for the (6,5) and (6,6) species. It should be noted that these values represent an estimated lower limit of the content ratio, as the small contribution of the $(2n, 2m)$ $S_{22}$ exciton absorption expected at almost the same photon energy is neglected. The content ratio of the emerged species is considered to be sufficiently high to not only modify the macroscopic optical properties of nanotube aggregates but also considerably modify their electric transport properties (see Methods and Supplementary Fig. 2).

### Coalescence of the enantiomers
Moreover, we examined the possibility of coalescence between the (6,5) and (5,6) nanotubes (equivalent to (11, −5)), which are

enantiomers of each other, because, in addition to the $(n, m) + (n, m) \rightarrow (2n, 2m)$ case confirmed above, $(n, m) + (m, n) \rightarrow (n + m, n + m)$ (Fig. 3d) may also be geometrically allowed. The results indicate that doubled (12,10) and (10,12) species are much more abundant than (11,11) species in the sample after heat treatment, as follows. We adjusted the nanotube dispersion to contain equal amounts of enantiomers of the (6,5) and (5,6) nanotubes (see Methods) and examined the possibility of generating (12,10) and (11,11) nanotubes. If the possibility of coalescence is perfectly equal among all possible cases, the probability of generating either (12,10) or (10,12) is equal to that of producing (11,11) nanotubes. The (12,10) and (10,12) nanotubes should show the same exciton absorption peak at the photon energy of 0.67 eV, and an exciton peak $M_{11}$ for (11,11) species of comparable intensity must also be observed at a photon energy of ≈1.8 eV[14,33]. Figure 3e, f shows the absorption spectrum after the reaction. As a result, the $M_{11}$ exciton absorption peak for (11,11) metallic nanotubes was not as clearly observed as that for (12,10) species in the optical absorption spectra. Although a small peak seemingly exists at around 1.7–1.8 eV, as indicated by the open triangles in Fig. 3f, this peak also exists before heat treatment; thus, it cannot be assigned solely to (11,11).

### Mechanism of the coalescence
Let us now discuss the mechanism of the efficient coalescence reaction that allowed for only nanotubes with relatively large chiral angles (i.e.,

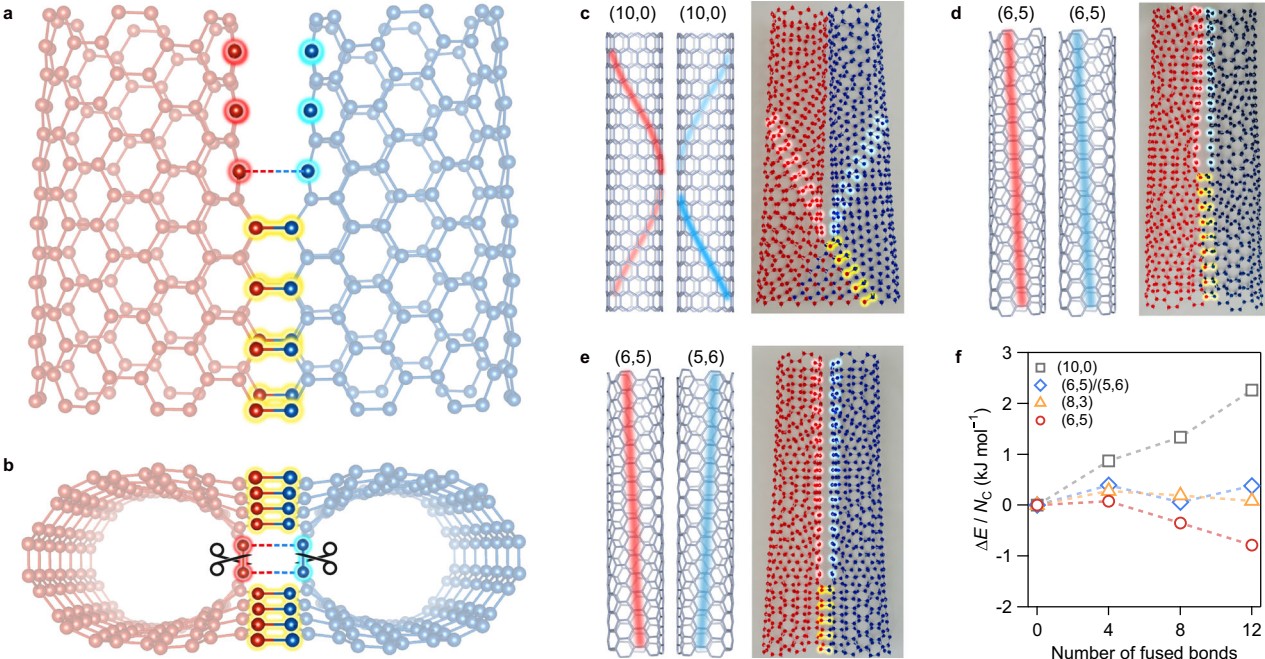

**Fig. 4 | Mechanism of coalescence with chiral-angle dependence.** Schematic of the sequential bond cleavage and recombination (SBCR) mechanism from the side (**a**) and the bottom (**b**). C-C bonds highlighted in red and blue are those that break and recombine. In particular, the red and blue highlighted in (**b**) will now break and recombine (broken lines). Schematics of the three-dimensional arrangement of the zigzag direction of two (10,0) nanotubes (**c**), two (6,5) nanotubes (**d**), and enantiomers of (6,5) and (5,6) nanotubes (**e**) contributing to cleavage and recombination for the coalescence (red and blue lines in each left panel). Each photograph on the right shows a coalescence reaction in progress demonstrated using a molecular structure model kit (MOL-TALOU®). **f** Energy difference $\Delta E$ of partially coalesced nanotubes relative to that of the initial two nanotubes normalized by the number of carbon atoms $N_C$ in the calculation, plotted as a function of the number of cleaved and newly formed bonds along the zigzag direction, as calculated using molecular mechanics simulations. Source data are provided as a Source Data file.

armchair and near-armchair types). According to previous theoretical studies on nanotube coalescence[36], from a geometric point of view, any $(n, m)$ structure is allowed to coalesce into the $(2n, 2m)$ structure, and the coalescence between $(n, m)$ and its enantiomer $(m, n)$ may also occur. However, the experimental results contradict this prediction. Therefore, there must be an unknown mechanism that drives the striking difference in the coalescence efficiency depending on the chiral angle. Here, we propose a sequential bond cleavage and recombination (SBCR) mechanism to understand the observed chiral-angle dependence of the coalescence efficiency (Fig. 4). In the SBCR scenario, once a few C−C bonds are connected between two $(n, m)$ nanotubes (yellow highlighted C−C bonds in Fig. 4a, b), the next candidate C−C bond to be cleaved and recombined is the one next to the just-connected bond (the red and blue highlighted C−C bonds (solid lines with scissors to be cleaved, and dashed lines to be recombined) in Fig. 4b) because of the highest strain on it among all C−C bonds due to stress concentration. The high strain on the next bond drives the cleavage of the bond in the original $(n, m)$ nanotubes and the formation of C−C bonds between the two $(n, m)$ nanotubes. The coalescence of the entire nanotube can be achieved if this reaction sequentially occurs as a chain reaction.

Figure 4c, d shows the geometry of the C−C bonds (red and blue highlighted C−C bonds) to be cleaved and recombined in the SBCR scenario for zigzag (10,0) and near-armchair (6,5) nanotubes, respectively. As shown in Fig. 4c, the distance between the candidate carbon atoms of the two (10,0) nanotubes that must be connected increases when the axes of the original nanotubes should be kept parallel (the right panel in Fig. 4c); this constraint is natural for the nanotubes in a bundle. The situation is similar between the enantiomers (6,5) and (5,6) (Fig. 4e). In contrast, those between two (6,5) nanotubes are much closer than those in the above cases, and the sequential reaction seems readily possible (Fig. 4d). Because of this geometrical difference, under

the SBCR scenario, only large-chiral-angle (near-armchair and armchair) $(n, m)$ nanotubes smoothly coalesce to form long $(2n, 2m)$ nanotubes. The cylindrical cross sections of the coalesced $(2n, 2m)$ structure and the two original tubes could not form perfect circles but became considerably distorted owing to the strain induced by the conversion junction among one $(2n, 2m)$ structure and two $(n, m)$ structures. The slight redshifts observed in the RBM of the original $(n, m)$ types after coalescence (Supplementary Fig. 7) can be attributed to circumferential stress resulting from cross-sectional distortion[37] in partially coalesced nanotubes.

Figure 4f shows the calculated energy change of partially coalesced nanotubes along the SBCR scenario for the coalescence reaction of three typical nanotube structures, (6,5) with $\theta = 27°$, (8,3) with $\theta = 15°$, and (10,0) with $\theta = 0°$, and (6,5) and its enantiomer counterpart (5,6), as a function of the number of cleaved and newly formed bonds along the zigzag direction (see Methods for details). For clarity, we plotted the results up to 12 fused bonds (see Supplementary Fig. 8 for the results up to all bonds combined, where all the $(2n, 2m)$ nanotubes have lower energy than the initial two $(n, m)$ nanotubes, ultimately). In the simulation, no constraint was imposed on the axis direction of the two original nanotubes for simplicity. The energy change required to increase the number of C−C bonds in the $(2n, 2m)$ structures indicates that the coalescence reaction of zigzag (10,0) nanotubes hardly occurs because of the increasing energy cost to form the next C−C bonds due to the high strain. There is no energy stabilization for the (8,3) and (6,5) + (5,6) cases either. In contrast, for the (6,5) case, after several C−C bonds are formed, the energy change due to the formation of one more C−C bond is negative; thus, the reaction could sequentially occur as a chain reaction. These results are consistent with the experimental results and support the proposed SBCR mechanism.

Finally, we discuss the effect of gas environment conditions on the coalescence reaction. When we placed a (6,5) nanotube membrane in a

high-vacuum-sealed quartz tube for heat treatment, interestingly, no signature of nanotube coalescence could be observed in the optical absorption and RBM Raman results even after heat treatment at 1000 °C (see Supplementary Fig. 9). The coalescence reaction occurred only when (6,5) nanotube membranes were placed in a relatively large quartz tube, where the vacuum level was maintained at ≈10⁻⁴ Pa, but a trace amount of residual and evolved gases similar to the composition of air was continuously supplied during heat treatment (see Methods and Supplementary Fig. 10). This result implies that a small number of residual gas molecules promotes the reaction[3]. Therefore, supplying a small amount of reactive gas molecules, such as oxygen, may enable them to play a role similar to electron irradiation during TEM observations[4], namely, breaking C–C bonds or removing C atoms to initiate and promote the nanotube coalescence reaction. To test this hypothesis, we conducted heat treatment experiments in an argon atmosphere (1 kPa) while controlling the partial pressure of oxygen gas at 10 Pa and below 10⁻⁴ Pa, respectively. To prevent excessive oxidation, the reaction temperature was set to 600 °C. Figure 5 compares the absorption spectra after heat treatment using oxygen gas (Fig. 5a) and that without oxygen gas (Fig. 5b). A distinct exciton absorption peak of (12,10) nanotubes appeared only in the 10 Pa oxygen condition. Raman spectroscopy further confirmed that nanotubes with twice the original diameter formed exclusively after reactions conducted in the presence of oxygen gas (see inset). These findings demonstrate that nanotube coalescence via heat treatment can occur efficiently even at much lower temperature (600 °C) than previously reported (≈1700 °C)[5] when an appropriate assisting gas is used under an appropriate condition. The previous theoretical study on the reaction of oxygen and nanotubes[17] suggests that oxygen atoms have strong energetic favorability to adsorb on small-diameter carbon nanotubes and form unzipped C-O-C epoxy chains along a direction of minimum angle to the tube axis. If this reaction occurs between two adjacent nanotubes with same chirality, it might help cleaving C-C bonds along the red and blue lines in each left panel in Fig. 4c–e, and preferentially promote the coalescence reaction via the SBCR mechanism. Further detailed mechanisms of the chemical reaction with oxygen and the optimal reaction conditions remain to be clarified in the future studies.

In conclusion, we demonstrated the efficient coalescence of $(n, m)$ carbon nanotubes into $(2n, 2m)$ nanotubes with doubled diameter and the same chiral angle via heat treatment. The coalesced nanotubes exhibited distinct exciton resonance peaks in the absorption spectra, indicating that a large number of coalesced nanotubes is generated all over the sample with an electronic structure and optical properties inherent to the $(2n, 2m)$ nanotubes. In addition to optical absorption spectroscopy, the doubled diameters of the coalesced nanotubes were confirmed via the TEM measurements and the emergence of the additional RBM peak at half the wavenumber of that of the original $(n, m)$ nanotubes. The preservation of the chiral angle was also directly observed in the TEM images. A distinct chiral-angle dependence of the coalescence efficiency was observed, and an SBCR mechanism was proposed based on the results. Furthermore, the presence of trace amounts of oxygen was found to enable the coalescence reaction even at more than 1000 °C lower temperature than previously reported. These findings provide routes for the structure-selective synthesis of large-diameter single-walled carbon nanotubes with well-defined chirality from structure-purified small-diameter nanotubes and for fabricating covalently connected macroscopic nanotube aggregates with modified optical properties, electric conductivity, thermal conductivity, and mechanical strength via postprocessing.

## Methods
### Fabrication of carbon nanotube membranes
Single-chirality carbon nanotube membranes were fabricated using a vacuum filtration method[32]. Single-chirality nanotubes with $(n, m) =$ (10,0), (9,1), (9,2), (8,3), (6,5), and (6,6) were separated from the starting materials of HiPco (NanoIntegris) and CoMoCAT (SG65 or SG65i, Sigma-Aldrich) samples using gel column chromatography[29,31]. The separated nanotubes were dispersed in pure water with various surfactants, including sodium deoxycholate (DOC), sodium cholate (SC), sodium dodecyl sulfate (SDS), and sodium lithocholate (LC). We prepared dispersions of (10,0) nanotubes (NanoIntegris HiPco, 0.12 μg mL⁻¹ in 0.3% SC + 0.9% SDS + 0.08% LC)[31], (9,1) nanotubes (NanoIntegris HiPco, 0.12 μg mL⁻¹ in 0.3% SC + 0.9% SDS + 0.06% LC)[31], (9,2) nanotubes (NanoIntegris HiPco, 0.09 μg mL⁻¹ in 0.3% SC + 0.9% SDS + 0.12% LC)[31], (8,3) nanotubes (NanoIntegrisHiPco, 0.12 μg mL⁻¹ in 0.3% SC + 0.9% SDS + 0.1% LC)[31], and (6,5) nanotubes (CoMoCAT SG65, 1.14 μg mL⁻¹ in 0.5% SC + 0.5% SDS + 0.03% DOC)[29]. The enantiomers, (5,6) (equivalent to (11, − 5)) and (6,5) nanotubes were prepared as described in Ref. 30. For the experiments using the enantiomers, we mixed (5,6) nanotubes (CoMoCAT SG65, 1.14 μg mL⁻¹ in 0.5% SC + 0.5% SDS + 0.028% DOC) and (6,5) nanotubes (CoMoCAT SG65, 1.21 μg mL⁻¹ in 0.5% SC + 0.5% SDS + 0.026% DOC) to control the enantiomeric ratio in the sample to 1:1. As for metallic (6,6) nanotubes, at first metallic nanotube mixture was obtained as unadsorbed fraction in 0.5% SC + 0.5% SDS elution[29], then the metallic nanotube fraction was adsorbed to the gel at 26 °C and pH 10.1 adjusted with NaOH, and the adsorbed metallic nanotubes were eluted by stepwise increase of DOC concentration (0.005% step in 0.5% SC and 0.5% SDS). The (6,6) nanotubes were obtained at 0.045% DOC (CoMoCAT SG65i, 2.4 μg mL⁻¹ in 0.5% SC + 0.5% SDS + 0.045% DOC). To confirm that the coalescence reaction occurs regardless of the original material, we also prepared (6,5) nanotubes from a different original material (Nopo HiPco)[29]. The optical absorption spectra indicating the coalescence of this material are shown in Supplementary Fig. 11. In the vacuum filtration process, a polycarbonate membrane filter with a pore size of 100 nm (MERCK, VCTP02500) and a filter holder with an effective filtration area of 2.1 cm² (ADVANTECH, KGS-25) were used. Each dispersed nanotube solution was diluted to below the critical micelle concentration of each surfactant (0.08%–0.25% (w/v) for DOC, 0.39%–0.65% (w/v) for SC, and 0.20%–0.29% (w/v) for SDS), and it was filtered at 50–80 kPa for ≈ 30 min. Following the nanotube solution, hot water (5 mL) was poured into the filtering system to remove excess surfactants. Then, the membrane filter with the nanotubes was dried in air at 1–3 kPa for 30 min. After cutting the obtained nanotube membrane on the filter to a size suitable for transfer onto sapphire substrates, it was immersed in chloroform for 15 min to dissolve the filter. The nanotube membrane floating on chloroform was scooped using a sapphire substrate and cleaned using chloroform, ethanol, and acetone in the same order.

### Coalescence reactions
Coalescence reactions were examined by heating single-chirality nanotube membranes on a sapphire substrate in a quartz tube using an electric furnace (Fig. 2b). For high temperature (900–1000 °C) heat treatments, the pressure in the quartz tube was reduced to ≈5.3 × 10⁻⁴ Pa. Initially, the furnace was heated to 300 °C and kept for 10 min to remove residual molecules other than nanotubes in the membrane. Then, the temperature of the furnace was increased to 900 or 1000 °C for 15 min for the coalescence reaction. For the low-temperature (600 °C) heat treatment (Fig. 5), the (6,5) nanotube membrane on sapphire substrate was heated to 600 °C and kept for 15 min in an Ar (99.9999% purity, (Fig. 5a) or containing 1% oxygen (Fig. 5b)) atmosphere controlled at 1 kPa in the quartz tube.

### Optical spectroscopy
Optical absorption spectra were measured using the combination of a Fourier-transform infrared spectrometer (JASCO, FT/IR-6600) and a UV spectrophotometer (SHIMADZU, UV-1800). Raman spectroscopy was conducted using a micro-Raman setup (Nanophoton, RAMAN-touch and Renishaw, inVia confocal Raman microscope). The excitation laser wavelengths for (6,5) and (6,6) species were 532, 488, and

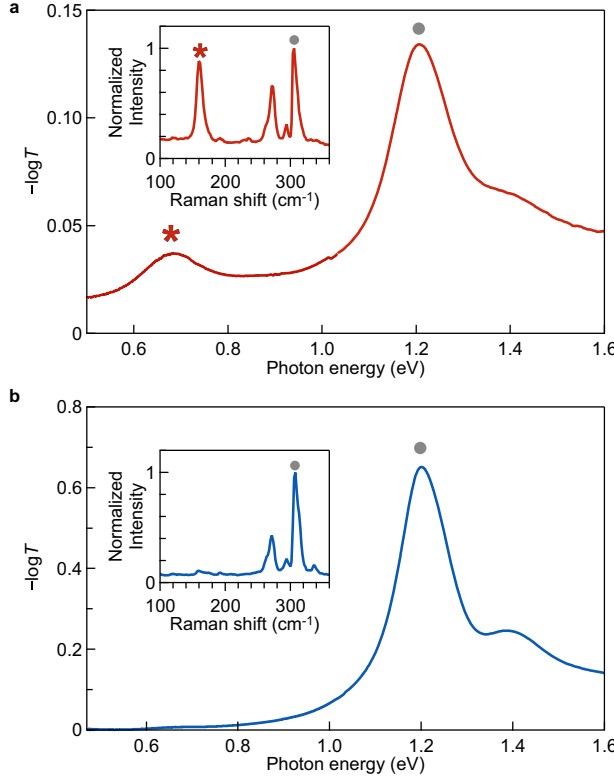

**Fig. 5 | Effect of the addition of oxygen gas.** Optical absorption ($-\log T$, where $T$ is transmittance) spectra of (6,5) nanotube membranes after heat treatment at 600 °C in a 1 kPa argon atmosphere (**a**) with (10 Pa) and (**b**) without (<10$^{-4}$ Pa) oxygen gas. The filled circles and asterisks indicate the lowest energy exciton peaks of (6,5) and (12,10) nanotubes, respectively. Insets in (**a**, **b**) are Raman spectra of the RBM. The filled circles and asterisks indicate the RBM peaks for (6,5) and (12,10) species, respectively. Source data are provided as a Source Data file.

785 nm, respectively. In the Raman spectra of (6,5) and (6,6) nanotubes, we observed not only the RBM peak of the major ($n$, $m$) nanotubes but also those of minority nanotubes included in the membrane owing to incomplete separation. This is because RBM observation relies on resonant Raman scattering[34], and even when a minor component is resonant to the excitation laser, its Raman signal is strongly enhanced.

## Electrical conductivity measurements

To examine the macroscopic changes in transport properties after the nanotube coalescence reaction throughout the membrane, we performed four-terminal electrical resistance measurements of (6,5) and (6,6) membranes before and after the reaction at 900 °C (Supplementary Fig. 2). The (6,5) or (6,6) nanotube membranes were prepared on 5 × 5 mm$^2$ sapphire substrates. Gold electrodes were deposited on the nanotube membranes, and their current–voltage characteristics were investigated via four-terminal sensing using a source meter (KEITHLEY, 2636B). We found that the direction of the change in resistance was opposite for semiconducting (6,5) and metallic (6,6) nanotubes. The resistance of semiconducting (6,5) nanotubes decreased after heat treatment (Supplementary Fig. 2a), whereas the resistance of metallic (6,6) nanotubes increased (Supplementary Fig. 2b). This striking difference between the semiconducting and metallic nanotubes indicates that the observed change was not merely a consequence of the degradation of the material. The change in conductivity might be attributed to sidewall functionalization or the

adsorption of residual gas molecules, including oxygen, which can induce p-type doping. Larger diameter nanotubes formed via coalescence have lower band gaps than the original ones and may be more readily doped. The existence of the small amount of tripled (3$n$, 3$m$) nanotubes that are always metallic regardless of the ($n$, $m$) of the original nanotubes may also affect the result. Although the detailed mechanism of the observed opposite trend is unclear at this stage, these results indicate that the coalescence reaction can be used to control the electrical transport properties of the membranes via postprocessing.

## Transmission electron microscopy

For TEM observations, thin nanotube membranes were prepared on a molybdenum grid. The membranes transferred on the grids were treated at 300 °C in a vacuum for 10 min to remove residual molecules on the nanotubes and heat-treated at 900 °C for 15 min for the coalescence reaction. TEM observation was carried out on an aberration-corrected TEM instrument (FEI Titan Cubed) at an acceleration voltage of 80 kV under $4 \times 10^{-6}$ Pa in the specimen column using a monochromator for the incident electron beam ($\Delta E = 0.15$ eV). Images were captured and processed on a CMOS camera (Gatan OneView, D mode, 4096 × 4096 pixels) operated in the binning 2 mode. The images were recorded at under-focus conditions (defocus value: ca −6 nm) at an electron dose rate of ca $1 \times 10^6$ e$^-$ nm$^{-2}$ s$^{-1}$, and the exposure time was set to 1.0 s. TEM image simulation was performed by using a multi-slice procedure implemented in Elbis software[38] using the experimental observation conditions described above. Diffraction patterns of individual nanotubes were obtained using the same microscope at 80 kV by forming a parallel electron beam with a diameter of ≈ 5 nm. Diffraction patterns were processed on DifPACK module of DigitalMicrograph software (Gatan, Inc.) to determine a chiral angle of an individual nanotube.

## Estimation of coalescence efficiency

Coalescence efficiencies were calculated based on the total area of the absorption of the first sub-band optical transition obtained through fitting procedures (see Supplementary Fig. 5). The absorption spectra were fitted with a sum of a linear baseline (gray solid line) and Lorentz functions for the first sub-band optical transition (red filled areas) and the phonon sideband (blue filled areas). Supplementary Fig. 6 displays the fitting procedure of the absorption spectrum with the exclusion of the background baseline.

## Molecular mechanics simulations

Avogadro (ver. 1.2.0)[39] was used to perform molecular mechanics calculations to evaluate changes in the total energy as a function of the number of connected C–C bonds between two ($n$, $m$) nanotubes. Calculations were performed for (10,0) (zigzag, length 38 Å), (8,3) (chiral, length 42 Å), (6,5) (near-armchair, length 42 Å), and (6,5) + (5,6) (near-armchair enantiomers, length 42 Å) nanotubes using MMFF94 and MMFF94s as force fields. According to the SBCR mechanism proposed in the main text, C–C bonds were cleaved and recombined along the zigzag direction from the center of the nanotube, and the energy for each step was obtained after structural relaxation.

## Q-mass analysis and gas composition

Residual and evolved gases during the coalescence reactions were detected using a quadrupole mass spectrometer (CANON ANELVA CORPORATION, M201QA-TDM), which was connected to a quartz tube in an electric furnace. The result is shown in Supplementary Fig. 10.

## Schematics

Schematics of carbon nanotubes, fullerenes, and their coalesced molecules were drawn using VESTA 3[40].

## Data availability

The data that support the findings of this study are available from the corresponding author upon request. Source data are provided with this paper.

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

## Acknowledgements

This work was supported by JST CREST Grant Number JPMJCR18I5 (Y.M.), JSPS KAKENHI Grant Numbers JP22K18287 (Y.M.), JP24H00044 (Y.M.), JP23H01791 (T.N.), JP23H04874 (K.H.), and JP24K08253 (O.C.), and JST FOREST Grant Number JPMJFR222N (T.N.). The TEM observations were supported by Kyoto University Nano Technology Hub in "Advanced Research Infrastructure for Materials and Nanotechnology in Japan (ARIM)" sponsored by the Ministry of Education, Culture, Sports, Science and Technology (MEXT), Japan. We thank Hiroki Kurata, Atsushi Yamaguchi, and Tsutomu Kiyomura for their assistance with TEM observations shown in Supplementary Fig. 3, and Koichi Okudaira for his assistance with TEM observations in the main text.

## Author contributions

Y.M. directed the project and wrote the manuscript. Y.M., A.T., and T.N. conceived the concept. A.T. prepared the nanotube membranes, carried out all the experiments and computational simulations except for the preparation of single chirality enriched nanotube samples and TEM observations. K.H. and O.C. contributed to TEM observations and simulations. T.T. and H.K. prepared single chirality enriched nanotube samples. T.N. supported optical measurements, figure preparation, and writing the manuscript. All authors contributed to the preparation of the manuscript.

## Competing interests

The authors declare no competing interests.
