## [Transparent Peer Review file · Nature Communications]

Coalescence of carbon nanotubes while preserving the chiral angles

Corresponding Author: Professor Yuhei Miyauchi

Version 0:

Reviewer comments:

Reviewer #1

(Remarks to the Author)

This paper shows the possibility to form $(2n,2m)$ SWNTs from (n,m) tubes via solid state chemical reaction at reduced pressure at about 1000 oC and with trace amount of oxidant like water or oxygen, when the chiral angle of the starting SWNTs is 30 degrees or close to 30 degrees. These results are novel and relevant to the field offering the possibility to produce large diameter mono-chiral SWNTs, which cannot be made via direct synthesis or solution separation. The mechanism well explaining the observed experimental results is proposed, based on molecular energy level modeling. Both experimental as well as modeling methods are solid, as well as data analysis, interpretation and conclusions. Methods are given in detail enough to allow reader to repeat the experiments.

I recommend paper to be published.

Reviewer #2

(Remarks to the Author)

The paper by Takahura et al. was very interesting to read. The paper is about heating two carbon nanotubes that are side by side and them joining to form a larger nanotube. This has been modeled in the past but experimentally has been an inefficient process seldomly observed. The paper reports that the efficiency of this conversion can be much higher if the two nanotubes are the same n,m chirality. Using chirality sorted samples, they observe the combination of 6,5 nanotubes into 12,10 nanotubes. As evidence they see larger diameter nanotubes in TEM and a new peak in Raman and absorption spectroscopy consistent with 12,10 nanotubes – but not necessarily proving their existence (see below). It is not clear if there would ever be widespread, high-impact utility for this conversion, but the types of nanotubes that the paper says do join compared to those that do not join provides important information about the chemistry and high temperature behaviors of sp^2 carbon materials. The paper presents a nice model of how C-C bonds break and reform that seems to support the experimental results. Even still, doubts linger in this paper that I think need to be thoroughly addressed.

1) There is no direct evidence of 12,10 nanotubes provided. Only TEM that shows that there are at least some CNTs about the right diameter and only optical absorption and Raman spectra that are consistent with 12,10. To more strongly identify the n,m , typically TEM diffraction data are needed that can quantify n,m . Alternatively, measurement of S22 and S11 combinations from PLE maps can be better used to evidence that 12,10 has formed.

2) One problem is that graphene nanoribbons (GNRs) can also be formed by heating and unzipping CNTs at high temperatures, sometimes in the presence of oxygen. There are many papers on this GNR synthesis approach. An unzipped 6,5 turned into a GNR will have a width of about $7.57 \text{ Angstroms} * 3.14159 = 23.8 \text{ Angstroms}$. The radial-like breathing mode of a GNR of this width is very close to 150 cm^{-1} , not that far from the new vibrational mode observed here. The radial-like breathing mode of a GNR also disperses with Raman laser energy. GNRs will also have exciton-like absorption and bandgaps $< 1 \text{ eV}$ in absorption. More work is needed to show that 6,5- derived GNRs are not responsible for the absorption and Raman spectra features (i.e., the macroscopic spectroscopic evidence) with only the very occasional larger diameter nanotube observed in TEM.

3) Oxygen is cited as a key ingredient. What about all the other components of air?

Reviewer #3

(Remarks to the Author)

Takakura et al. describe a study where heat treatment of thin films of SWCNTs results in coalescence of some fraction of the (n,m) SWCNTs into (2n, 2m) SWCNTs. This is a very interesting study and set of results, which (if optimized) could add to the available methods for attaining larger diameter near-monochiral SWCNT species. As such, I think it could potentially be publishable in Nature Communications, but the authors should attempt to address several comments below. Specifically, some additional strategic characterization could strengthen the conclusions, and some slightly more careful control over the synthesis conditions could help to improve the efficacy of the coalescence reaction, both of which could make this method quite useful to the community.

- Can the authors measure emission spectra of the films before and after coalescence? This would help to confirm e.g. that the coalescence of (6,5) SWCNTs results in predominantly (12, 10) SWCNTs.
- In the authors' analysis of the coalescence efficiency, I cannot tell if they are taking into account the absorption of the S22 peak of the large-diameter coalesced SWCNT, e.g. for the (12, 10) SWCNT in the (6,5) coalescence. Does this introduce error into the calculation, and if so, approximately how much? Also, is there any way in which to analyze the absorbance spectra, e.g. by taking first or second derivative spectra, to discriminate the S22 of the coalesced tube from the S11 of the original tube? Would the peaks be narrow enough at low temperature to resolve via derivative spectra?
- The role of oxygen in catalyzing the current synthesis is unclear. The authors show results for a single partial pressure of oxygen that is established solely by the vacuum achieved by the particular setup they are using. A more concise method would be to flow specific partial pressures of oxygen (in an inert carrier gas, e.g. argon) into a setup that can achieve lower pressure. Do the authors have a way of trying this? That way, the partial pressure of oxygen could be tuned substantially and they might even get better coalescence efficiency.
- The authors note that the (6,5) absorption intensity decreases, indicating that (6,5) SWCNTs are coalescing into (12,10) SWCNTs. This appears to be the case, but it also appears that the (6,5) SWCNTs are being consumed by e.g. oxidation by the trace amount of oxygen in the reaction vessel. Can the authors estimate the percentage of original (6,5) SWCNTs that coalesce versus disappear via oxidation?
- In Fig. S7, what is the reason for the red shift of the (6,5) RBM mode after heat treatment?
- Since the heat-treated samples that coalesce are treated in an oxygen-containing environment, are the authors confident that there is no oxygen sidewall functionalization? Or is it possible that the resulting SWCNTs are p-type doped because of oxygen adsorption? S11 reduction and broadening can occur from this p-type doping and this can typically be observed by a decrease in the ratio of S11 to S22 absorption intensity. The authors should show absorption spectra over a wider range so that the relative S11:S2 absorption ratio can be observed before and after heat treatment. This is an important consideration, because any p-type oxygen doping would increase the conductivity of the SWCNT film and could explain the conductivity change shown in Fig. S1.

Reviewer #4

(Remarks to the Author)

This paper reports a chirality preserved coalescence between carbon nanotubes under heat treatment. A trace amount of oxygen is found to determine the reaction rate. The author's finding echoes the results in previous papers that thermal treatment leading to the coalescence between 2 nanotubes with the same chirality. This paper, however, provides further results on how chirality could influence the yield of doubled nanotube. The result of this paper is solid with enough detail provided in methods for the work to be reproduced, while several points need to be further clarified:

1. Oxygen has long been known to help the coalescence between CNTs, but also creates defects. The authors are encouraged to provide characterizations on the defect levels. In addition, the authors should clarify how oxygen 'cuts C-C bonds or remove some C atoms', and how oxygen leaves (or remains in) the CNT once the bonds are cut.
2. The energy calculation Fig 4 f needs further clarification. Is it the energy difference between the CNT before and after the coalescence (of a unit cell in CNTs), or is it the reaction energy barrier (for the CC bond breaking and else)? If it refers to the energy barrier, the author should provide a list of reaction paths that have been compared to calculate the energy barrier in this paper. Else if it refers to the energy difference (before and after coalescence), it seems that the coalescence of (10,0) CNT is thermally (but not only kinetically) unfavourable. (Since the ΔE for each step of the coalescence is positive, the total ΔE would also be positive.) This indicates that the larger diameter (m,0) CNT is, the less thermally favourable it is, which is confusing.

Version 1:

Reviewer comments:

Reviewer #2

(Remarks to the Author)

The new TEM data, edits, and responses alleviate my previous concerns. I support publication of this manuscript.

Reviewer #3

(Remarks to the Author)

The authors have addressed the reviewer comments sufficiently. I am glad that the suggested O₂/Ar experiment allowed the authors to demonstrate that the coalescence can occur at even lower temperatures than prior studies. This is a nice improvement to the study.

Response Letter

Dear Editor,

Thank you for handling our manuscript NCOMMS-24-25705 entitled “Efficient chiral-angle-preserved coalescence of carbon nanotubes.” We have conducted additional experiments and carefully revised the manuscript using the constructive and helpful reviewers’ comments. Your consideration of the revised manuscript is greatly appreciated. Below we present a summary of changes in the manuscript.

Summary of changes

- On page 1, we added authors who conducted additional TEM observations.
- In the abstract (4th line from bottom), we added a sentence.
- On page 3 (2nd line from bottom), we added a sentence.
- On page 4 (8th line from bottom), we added a sentence according to Q3-6).
- On page 5 (2nd line), we added a sentence according to Q3-2).
- On page 5 (2nd line from bottom), we added sentences according to Q2-1).
- On page 7 (6th line from bottom of 1st paragraph), we add a sentence according to Q3-2).
- On page 9 (4th line from bottom of 2nd paragraph), we add a sentence according to Q3-5).
- On page 9 (bottom line), we add a sentence according to Q4-2).
- On page 10 (5th line from bottom), we added sentences according to Q2-3) and Q3-3).
- On page 11 (7th line from bottom of 1st paragraph), we added sentences according to Q4-1).
- On page 11 (2nd line from bottom), we added a sentence.
- On page 13 (bottom line), we added a sentence to explain the additional experiment.
- On page 14 (bottom line), we added sentences about the additional TEM measurements.
- On page 19, we added a reference (Ref. 38).
- On page 19, in Acknowledgements and Author contributions, we added sentences.
- On page 22 (Fig. 2), we added panels of Figs. 2d, 2e, and 2f and their captions according to Q2-1).
- On page 24 (Fig. 4), we modified caption according to Q4-2).
- On page 25, we added Fig. 5 according to Q2-3).
- In Supplementary Note 2 (8th line), we added sentences according to Q3-6).
- In Supplementary Information, we added Supplementary Fig. 1 according to Q3-6).
- In Supplementary Information, we added Supplementary Fig. 8 according to Q4-2).
- Some figures in the previous manuscript were moved to Supplementary Fig. 3 and Supplementary Fig. 9.

Responses to the Reviewers' comments and suggestions

We are grateful for the reviewers' comments and suggestions for improving the manuscript. Using the reviewers' insightful comments, we have revised the manuscript. In the following, point-by-point responses to the reviewers' comments and the revisions included in the revised manuscript are presented.

For Reviewer #1 (Remarks to the Author):

This paper shows the possibility to form (2n,2m) SWNTs from (n,m) tubes via solid state chemical reaction at reduced pressure at about 1000 °C and with trace amount of oxidant like water or oxygen, when the chiral angle of the starting SWNTs is 30 degrees or close to 30 degrees. These results are novel and relevant to the field offering the possibility to produce large diameter mono-chiral SWNTs, which cannot be made via direct synthesis or solution separation. The mechanism well explaining the observed experimental results is proposed, based on molecular energy level modeling. Both experimental as well as modeling methods are solid, as well as data analysis, interpretation and conclusions. Methods are given in detail enough to allow reader to repeat the experiments.

I recommend paper to be published.

We thank the reviewer for his/her recommendation for publication of our paper.

Reviewer #2 (Remarks to the Author):

The paper by Takahura et al. was very interesting to read. The paper is about heating two carbon nanotubes that are side by side and them joining to form a larger nanotube. This has been modeled in the past but experimentally has been an inefficient process seldomly observed. The paper reports that the efficiency of this conversion can be much higher if the two nanotubes are the same n,m chirality. Using chirality sorted samples, they observe the combination of 6,5 nanotubes into 12,10 nanotubes. As evidence they see larger diameter nanotubes in TEM and a new peak in Raman and absorption spectroscopy consistent with 12,10 nanotubes – but not necessarily proving their existence (see below). It is not clear if there would ever be widespread, high-impact utility for this conversion, but the types of nanotubes that the paper says do join compared to those that do not join provides important

information about the chemistry and high temperature behaviors of sp² carbon materials. The paper presents a nice model of how C-C bonds break and reform that seems to support the experimental results. Even still, doubts linger in this paper that I think need to be thoroughly addressed.

We thank the reviewer for his/her evaluation of our paper as “very interesting”. According to the reviewer’s insightful comments, we conducted additional experiments, and improved the manuscript. The point-by-point responses to the reviewer’s concerns are shown below.

Q2-1) There is no direct evidence of 12,10 nanotubes provided. Only TEM that shows that there are at least some CNTs about the right diameter and only optical absorption and Raman spectra that are consistent with 12,10. To more strongly identify the n,m, typically TEM diffraction data are needed that can quantify n,m. Alternatively, measurement of S22 and S11 combinations from PLE maps can be better used to evidence that 12,10 has formed.

We thank the reviewer for this important advice. We agree that no direct evidence of (12,10) nanotubes were shown in the previous manuscript and more direct data should be shown. In Fig. 2 of the revised manuscript, we included newly obtained TEM images of many CNTs with the diameter consistent with that of (12,10), and high-resolution TEM image with a moiré pattern consistent with that expected for (12,10). We also added a nanobeam diffraction data that suggests the newly formed CNT’s chiral angle is $\sim 27^\circ$. Based on these new TEM observations, the chiral indices of the generated CNTs were reliably identified as (12,10) CNTs. These new data were added as **Figures 2d, 2e, and 2f** of the revised manuscript, and related sentences were added **at pages 5 to 6**, as **“In contrast, in the membrane after heat treatment (Fig. 2d), we observed... into a single nanotube with triple the diameter can also occur.”**.

With regard to the PLE maps, we could not add it this time because it was difficult to measure PL from the generated CNTs in the membrane (the detailed reason is described in the response to the 3rd reviewer Q3-1). However, now we are confident that the newly added TEM observations provide sufficient evidence of (12,10) CNTs.

Q2-2) One problem is that graphene nanoribbons (GNRs) can also be formed by heating and unzipping CNTs at high temperatures, sometimes in the presence of oxygen. There are many papers on this GNR synthesis approach. An unzipped 6,5 turned into a GNR will have a width of about $7.57 \text{ \AA} \times 3.14159 = 23.8 \text{ \AA}$. The radial-like breathing mode of a GNR of this width is very close to 150 cm^{-1} , not that far from the new vibrational mode observed here. The radial-like breathing mode of a GNR also disperses with Raman laser energy. GNRs will also have exciton-like

absorption and bandgaps < 1 eV in absorption. More work is needed to show that 6,5- derived GNRs are not responsible for the absorption and Raman spectra features (i.e., the macroscopic spectroscopic evidence) with only the very occasional larger diameter nanotube observed in TEM.

The possibility that the observed nanomaterials are GNRs can be safely ruled out by the difference of the expected TEM images of the (12,10) CNT and a GNR with the similar width as shown in Fig. R1 below. TEM image of CNT should have two dark parallel lines, but the GNR should not. We never observed GNR-like material in the TEM images. Moreover, although the apparent width of GNRs should be dependent on the angle of the GNR plane against the TEM focal plane, we always observed the two dark parallel lines with discretized widths (mostly twice (or triple) the diameter of the original (6,5) nanotubes) as shown in Fig. 2d, which supports that the observed material has a cylindrical shape. Therefore, we can conclude that the materials generated after the heat treatment were CNTs.

Fig. R1. Simulated TEM images of (top) (12,10) nanotube and (bottom) graphene nanoribbon.

Q2-3) Oxygen is cited as a key ingredient. What about all the other components of air?

In order to examine the effect of the presence of oxygen more clearly, we conducted additional experiment with controlled partial pressure of oxygen gas (10 Pa) in argon atmosphere of 1 kPa at relatively low temperature of 600 °C. Consequently, the coalescence could occur only under the presence of oxygen gas. The results are added as Fig. 5 in the revised manuscript. Corresponding discussion is also added at the end of the discussion part (at pages 10 to 11). These results not only suggest that the presence of oxygen gas promotes nanotube coalescence, but also indicates that, under appropriate conditions, the coalescence reaction can occur even at much lower temperature (600 °C) than previously reported (1700 °C), which further raises the usefulness and impact of this study.

With regard to contribution of other potentially reactive air components at high temperatures such as H₂O and CO₂, we have not yet addressed. However, within the time given to us to revise the manuscript,

we were only able to perform the above experiment on oxygen gas. The 1st author who mainly conducted the experiments has already left the lab, and the possible number of additional experiments was limited owing to the limitation of the remaining amount of the purified (6,5) CNTs of the same batch used to other experiments compared in this study. Therefore, we would like to leave the effects of various gases other than oxygen for initiating or promoting the coalescence as a topic for future research.

Considering that we only studied the effect of oxygen, in the revised manuscript, we only claim the fact that addition of trace amounts of oxygen gas promotes the coalescence reaction even at relatively low temperature of 600 °C. Accordingly, the discussion on the effect of oxygen gas has been moved to the final part of the discussion (at pages 10 (2nd paragraph) to 11) just before the Conclusion part.

Reviewer #3 (Remarks to the Author):

Takakura et al. describe a study where heat treatment of thin films of SWCNTs results in coalescence of some fraction of the (n,m) SWCNTs into (2n, 2m) SWCNTs. This is a very interesting study and set of results, which (if optimized) could add to the available methods for attaining larger diameter near-monochiral SWCNT species. As such, I think it could potentially be publishable in Nature Communications, but the authors should attempt to address several comments below. Specifically, some additional strategic characterization could strengthen the conclusions, and some slightly more careful control over the synthesis conditions could help to improve the efficacy of the coalescence reaction, both of which could make this method quite useful to the community.

We thank the reviewer for his/her evaluation of our paper as “very interesting” and “potentially be publishable in Nature Communications”. Using the reviewer’s insightful comments, we have attempted additional experiments, and improved the manuscript. The point-by-point responses to the reviewer’s comments are shown below.

Q3-1) Can the authors measure emission spectra of the films before and after coalescence? This would help to confirm e.g. that the coalescence of (6,5) SWCNTs results in predominantly (12, 10) SWCNTs.

We thank the reviewer for this advice. According to his/her suggestion, we attempted to measure photoluminescence (PL) from the (12,10) CNTs after coalescence. However, meaningful PL signal could not be obtained around the photon energy of ~0.6–0.65 eV (wavelength range of 1900–2000 nm) where the PL of the (12,10) CNTs is expected. Since it is clear that (12,10) CNTs were synthesized

from the new results of TEM measurements (Figs. 2d–f), one possible reason why we were unable to measure PL is that the PL intensity is very weak and our experimental equipment may not have been sensitive enough. Indeed, it is well known that even the PL of the (6,5) CNTs are very weak when it is in the aggregated condition in the membrane, because of reduction of exciton oscillator strength due to aggregation, and/or the very fast non-radiative energy transfer to the small number of impurity low-gap CNTs included in the membrane. Moreover, because the expected photon energy of the PL from (12,10) CNTs (0.65 eV) are out of the detection range of our high-sensitive InGaAs detector (detectable up to ~1600 nm, 0.775 eV), we had to use less-sensitive IR-extended InGaAs detector which does not allow long time integration of very weak PL signals.

In addition, after the heat treatment, we observed not only the twice-diameter (12,10) species, but also less amount of triple-diameter species in the TEM images (Fig. 2d and Supplementary Fig. 3b). The triple-diameter nanotubes were a minor component. However, since they have lower gap than (12,10) CNTs, even a small amount of the tripled ones are included in a bundle, it is expected that the exciton energy transfer to the lower gap CNTs will occur and the PL of the doubled species will be quenched.

Q3-2) In the authors' analysis of the coalescence efficiency, I cannot tell if they are taking into account the absorption of the S22 peak of the large-diameter coalesced SWCNT, e.g. for the (12, 10) SWCNT in the (6,5) coalescence. Does this introduce error into the calculation, and if so, approximately how much? Also, is there any way in which to analyze the absorbance spectra, e.g. by taking first or second derivative spectra, to discriminate the S22 of the coalesced tube from the S11 of the original tube? Would the peaks be narrow enough at low temperature to resolve via derivative spectra?

In the evaluation of the efficiency shown in Fig. 3c, defined as $\eta_{nm} = A_{2n2m}/A_{nm}^{\text{before}}$, we only used the integrated absorption intensity of the S_{11} of the (n, m) CNTs *before* the heat treatment (A_{nm}^{before}) and the newly emerged S_{11} absorption of the $(2n, 2m)$ CNTs (A_{2n2m}). Therefore, the absorption of the S_{22} peak of the large-diameter coalesced CNTs does not affect the evaluation. However, for the calculation of the content ratio, defined as $A_{2n2m}/(A_{nm}^{\text{after}} + A_{2n2m})$, the S_{22} absorption of $(2n, 2m)$ CNTs potentially affects the value of A_{nm}^{after} , as the reviewer concerns. According to the Kataura plot, the expected contribution of the $(2n, 2m)$ S_{22} exciton absorption is at almost the same photon energy with the S_{11} exciton absorption of the original (n, m) CNTs (we added related a sentence “**while their S_{22} exciton absorption peak energies should be nearly the same as the S_{11} peak energies of the original (n, m) species.**” in the 1st paragraph of the page 5). Therefore, after the reaction, these peaks are expected to be overlapped, and cannot be distinguished from each other. This also prohibits discriminating the S_{22} absorption of the coalesced $(2n, 2m)$ CNTs from the overlapped S_{11} absorption of the remaining (n, m) CNTs by simple numerical analysis such as taking derivative of the spectra. In order to mention the above limitation in the evaluation of the content ratio, we added a sentence “**It should be noted that**

these values represent an estimated lower limit of the content ratio, as the small contribution of the (2n, 2m) S₂₂ exciton absorption expected at almost the same photon energy is neglected.” at the 1st paragraph in page 7.

With regard to the low temperature absorption spectra, it is difficult for us to add it within the reasonable time because we do not have a cryo-absorption measurement system. Generally speaking, exciton absorption line widths of the CNTs in the aggregation state are considerably broadened due to their inhomogeneous environment. Therefore, single nanotube spectroscopy would be required to clearly observe the decrease of the homogeneous line width at low temperature.

Q3-3) The role of oxygen in catalyzing the current synthesis is unclear. The authors show results for a single partial pressure of oxygen that is established solely by the vacuum achieved by the particular setup they are using. A more concise method would be to flow specific partial pressures of oxygen (in an inert carrier gas, e.g. argon) into a setup that can achieve lower pressure. Do the authors have a way of trying this? That way, the partial pressure of oxygen could be tuned substantially and they might even get better coalescence efficiency.

We thank the reviewer for this important advice. According to the reviewer’s advice, we conducted additional experiment with controlled partial pressure of oxygen gas (10 Pa) in argon atmosphere of 1 kPa at relatively low temperature of 600 °C. The experiment was a great success as shown in Fig. 5 in the revised manuscript. Corresponding discussion is also added at the end of the discussion part (from the 2nd paragraph of page 10). These results strongly confirm that the presence of oxygen gas promotes nanotube coalescence. Although we could not conduct many experiments under various oxygen partial pressure within the time allowed for the revision, this new finding further raises the impact of this study.

Q3-4) The authors note that the (6,5) absorption intensity decreases, indicating that (6,5) SWCNTs are coalescing into (12,10) SWCNTs. This appears to be the case, but it also appears that the (6,5) SWCNTs are being consumed by e.g. oxidation by the trace amount of oxygen in the reaction vessel. Can the authors estimate the percentage of original (6,5) SWCNTs that coalesce versus disappear via oxidation?

From the definition, the conversion efficiency shown in Fig. 3c provides the proportion of the carbon (C) atoms contained in the original CNTs that were incorporated into the coalesced CNTs. On the other hand, the remaining C-atoms in the (6,5) CNTs after the coalescence was ~30% as deduced from the absorption spectra before and after the heat treatment shown in Supplementary Fig. 4. Therefore, in

total, ~70% of the C-atoms in the original (6,5) CNTs are converted to the (12,10) CNTs or disappeared via oxidation. Therefore, the percentage of the C-atoms in the original (6,5) CNTs that involved in coalescence are roughly ~10% as shown in Fig. 3c, while the disappeared C-atoms via oxidation are ~60%.

Q3-5) *In Fig. S7, what is the reason for the red shift of the (6,5) RBM mode after heat treatment?*

This can be attributed to the circumferential stress [Ref. 37, Chang, T. Radial breathing mode frequency of single-walled carbon nanotubes under strain. Appl. Phys. Lett. 93, 061901 (2008)] in partially coalesced CNTs. We added the related description at the end of the 2nd paragraph in page 9 as “The slight red-shifts observed in the RBM of the original (*n, m*) types after coalescence (Supplementary Fig. 7) can be attributed to circumferential stress resulting from cross-sectional distortion³⁷ in partially coalesced nanotubes.”.

Q3-6) *Since the heat-treated samples that coalesce are treated in an oxygen-containing environment, are the authors confident that there is no oxygen sidewall functionalization? Or is it possible that the resulting SWCNTs are p-type doped because of oxygen adsorption? S11 reduction and broadening can occur from this p-type doping and this can typically be observed by a decrease in the ratio of S11 to S22 absorption intensity. The authors should show absorption spectra over a wider range so that the relative S11:S2 absorption ratio can be observed before and after heat treatment. This is an important consideration, because any p-type oxygen doping would increase the conductivity of the SWCNT film and could explain the conductivity change shown in Fig. S1.*

We measured the absorption spectra up to 3.0 eV for the coalescence of (6,5) CNTs (Fig. R2 below). There seems slight but not obvious change of the ratio before and after the reaction as shown in Fig. R2 (the peak intensity ratio ($S_{11} : S_{22}$) was 1:0.50 for “before”, and 1:0.55 for “after”).

Figure R2.

The increased conductivity of the (6,5) CNT film after coalescence may also be caused by the generation of the small amount of tripled ($3n, 3m$) CNTs that are metallic regardless of the (n, m) of the original CNTs. However, at this stage, it is not straightforward to specify one of the possibilities as the major origin of the conductivity change. Therefore, in the revised manuscript, we limit our discussion only to mention those possibilities. Accordingly, we added sentences “**The change in conductivity might be attributed to sidewall functionalization or the adsorption of residual gas molecules, including oxygen, which can induce p-type doping. Larger diameter nanotubes formed via coalescence have lower band gaps than the original ones and may be more readily doped. The existence of the small amount of tripled ($3n, 3m$) nanotubes that are always metallic regardless of the (n, m) of the original nanotubes may also affect the result.**”, in Supplementary Note 2. The broad band absorption data shown in Fig. R2 has been included Supplementary Fig. 1 in the revised Supplementary Information.

Reviewer #4 (Remarks to the Author):

This paper report a chirality preserved coalescence between carbon nanotubes under heat treatment. A trace amount of oxygen is found to determine the reaction rate. The author's finding echo's the results in previous papers that thermal treatment leading to the coalescence between 2 nanotubes with the same chirality. This paper, however, provide further results on how chirality could influence the yield of doubled nanotube. The result of this paper is solid with enough detail proceed in methods for the work to be reproduced, while several points need to be further clarified:

We thank the reviewer for evaluating the results of the paper solid. Using the reviewer's insightful comments, we improved the manuscript. The point-by-point responses to the reviewer's comments are shown below.

Q4-1) Oxygen has long been known to help the coalescence between CNTs, but also creates defects. The authors are encouraged to provide characterizations on the defect levels. In addition, the authors should clarify how oxygen 'cuts C-C bonds or remove some C atoms', and how oxygen leaves (or remains in) the CNT once the bond are cut.

The common indicator for the evaluation of the defect density in CNTs is G/D ratio of Raman spectra. Therefore, we analyzed the Raman spectra of G- and D-modes before and after the coalescence in Fig. R3 below. After the heat treatment, G/D ratio considerably increased (from 6 (before) to 18 (after)); therefore, the net defect density in the membrane decreased after heat treatment. This can be understood as the result of most of the defective (6,5) CNTs in the original membrane either being totally burnt or coalesced into (12,10) CNTs with less defects.

Fig. R3.

According to the newly added experimental results in Fig. 5, the defects created by oxygen may have special characteristics that are advantageous for promoting coalescence. Although it is difficult to further clarify the details of the defect structures generated during the coalescence reactions experimentally by our current capabilities, previous theoretical study on the reaction of oxygen and CNTs [Y. Guo, L. Jiang, and W. Guo, *Physical Review B* 82, 115440 (2010)] suggests that oxygen atoms have strong energetic favorability to adsorb on small-diameter CNTs and form unzipped C-O-C epoxy chains along a direction of minimum angle to the tube axis. If this reaction occurs between two adjacent CNTs with same chirality, it may help cleaving a few C-C bonds between the CNTs and could preferentially initiate the coalescence reaction via the proposed SBCR mechanism.

Therefore, we added a discussion on the above hypothetical mechanism about the contribution of oxygen for cutting the C-C bonds that could be reasonably inferred according to the previous theoretical study [Y. Guo, L. Jiang, and W. Guo, *Physical Review B* 82, 115440 (2010), added as Ref. 17] at 7th line from bottom of the 1st paragraph of page 11 as “The previous theoretical study on the reaction of oxygen and nanotubes¹⁷ suggests that ... initiate the coalescence reaction via the SBCR mechanism.”.

As the reviewer commented, if we could clarify the details of the oxygen-nanotube reactions during coalescence reaction, the paper will be further improved. However, it is difficult for us to further clarify the details of the reaction within the reasonable time allowed for the revision on our current experimental and theoretical capability. To indicate the existence of the remaining issues that should be clarified, we added a sentence at the end of the first paragraph of page 11 as “Further detailed mechanisms of the chemical reaction with oxygen and the optimal reaction conditions remain to be clarified in the future studies.”.

Q4-2) *The energy calculation Fig4f need further clarification. Is it the energy different between the CNT before and after the coalescence (of a unit cell in CNTs), or it is the reaction energy barrier (for the CC bond breaking and else)? If it refers to the energy barrier, the author should provide a list of reaction path that have been compared to calculate the energy barrier in this paper. Else if it refers to the energy difference (before and after coalescence), it seems that the coalescence of (10,0) CNT is thermal dynamically (but not only kinetically) unfavourable. (Since the delta E for each step of the coalescence is positive, the total delta E would also be positive.) This indicate that the larger diameter a (m,0) CNT is, the less thermal dynamically favourable it is, which is confusing.*

We are sorry that our previous manuscript was unclear on what we calculated and somewhat confusing. It is the former one, namely, the energy difference between the CNTs before and after (during) the coalescence. The increase of the energy for the (10,0) CNT is because the strain energy when a small number of C-C bonds are formed between two (10,0) CNTs is quite large in the proposed SBCR scenario. Therefore, after all the C-C bonds are connected and the cylindrical cross section of the larger diameter (20,0) CNT is recovered, the total energy per C-atom in the (20,0) CNT becomes smaller than the original (10,0) CNT. To avoid confusion, we added the calculation results for larger number of fused bonds in **Supplementary Fig. 8** in the revised manuscript, and added at **the end of page 9** as **“For clarity, we plotted the results up to 12 fused bonds (see Supplementary Fig. 8 for the results up to all bonds combined, where all the (2n, 2m) nanotubes have lower energy than the initial two (n, m) nanotubes, ultimately).”**. We also removed the term “energy barrier” that was inappropriate, and we replaced it to “energy difference” or “energy change” in the revised manuscript.